

# Making sense of variation in sclerochronological stable isotope profiles of mollusks and fish otoliths from the early Eocene southern North Sea Basin

Johan Vellekoop[1,2], Daan Vanhove[1,2], Inge Jelu[1,2], Philippe Claeys[3],Linda C. Ivany[4], Niels J. De Winter[5],
Robert P. Speijer[1], Etienne Steurbaut [2,1]

[1]Department of Earth and Environmental Sciences, KU Leuven, 3001 Heverlee, Belgium
[2]OD Earth and History of Life, Institute of Natural Sciences, 1000 Brussels, Belgium
[3]Archaeology, Environmental changes and Geochemistry Research Unit, Vrije Universiteit Brussel, 1050 Brussels, Belgium
[4] Department of Earth and Environmental Sciences, Heroy Geology Laboratory, Syracuse University, Syracuse  NY 13244-
1070, USA
[5] Faculty of Science, Vrije Universiteit Amsterdam, 1081 HV Amsterdam, Netherlands

*Correspondence to*: Johan Vellekoop (johan.vellekoop@kuleuven.be)

**Abstract.** Stable isotope sclerochemistry of mollusks and otoliths is frequently used for the reconstruction of paleotemperature and seasonality. Constraints on the paleoecology and –environment of these organisms, and how these factors influence intra- and inter-taxon isotope variability and variation, are thus highly valuable. We measured seasonal changes in $\delta^{18}O$ and $\delta^{13}C$ compositions in multiple specimens of two carditid bivalve species, a turritelline gastropod species, and two species of otoliths from demersal fish, from two early Eocene (latest Ypresian, 49.2 Ma) coquinas in the inner neritic Aalter Sand Formation, located in the Belgian part of the southern North Sea Basin (paleolatitude ~41°N). Results demonstrate variability among taxa in average, amplitude and shape of intra-annual $\delta^{18}O$ and $\delta^{13}C$ values. This intertaxon variability is at least partly caused by growth cessation during winters in turritellines and otoliths, leading to an incomplete representation of the seasonal cycle in their growth increments, compared to carditid bivalves. Other contributing factors to isotopic variability include sedimentary transport, mobility, and the lifespan of the specimens. Specifically, ophidiid fish otolith isotope records appear to reflect environmental conditions over a wider range of habitats and environments, due to sedimentary transport and postmortem transport by free-swimming predatory fish. Our study therefore highlights the variability between different taxa and environments in the shallow marine realm, which has implications for seasonality reconstructions. We show that by studying multiple taxa and specimens in a death assemblage, a more complete spectrum of isotope variation and variability becomes apparent.

## 1 Introduction

Reconstructions of past warm climates deliver essential information on the response of Earth's climate system to warming (Burke et al., 2018; Tierney et al., 2020). Datasets of past climate help us to assess the ability of climate models to simulate



warm climate scenarios, aiding in the improvement of future climate projections (Calvin et al., 2023) . Short-term archives of past climate, such as the incrementally mineralized parts of mollusks (shells) and fish (otoliths) crucially enable us to reconstruct climate variability on the seasonal timescale (Ivany, 2012). Oxygen isotope sclerochemistry has become a widely
applied tool for the reconstruction of intra-annual variation in ambient temperature and water composition (e.g. De Winter et al., 2018, 2020; Clark et al., 2022; Ivany and Judd, 2022). Carbon isotope composition of accretionary archives is explored to a lesser degree but contributes to our understanding of species metabolism and diet, and variations in primary productivity and dissolved inorganic carbon (DIC) compounds in water (e.g. McConnaughey and Gillikin, 2008; Van Horebeek et al., 2021; Clark et al., 2022).

For seasonal-scale climate reconstructions of the high-$CO_2$ Paleogene greenhouse world, considered a 'worst case scenario' for our anthropogenic global warming (Tierney et al., 2020), commonly used fossil groups include turritellid gastropods (e.g. Andreasson and Schmitz, 1996, 1998, 2000; Kobashi et al., 2001, 2004; Ivany et al., 2018) and carditid bivalves (e.g. Purton and Brasier, 1999; Kobashi et al., 2001, 2004; Ivany et al., 2004; Keating-Bitonti et al., 2011; Vanhove et al., 2012; Sessa et al., 2012). In addition, fossil otoliths are also regularly utilized, in particular those from non-migratory
groundfish such as ophidiids (Vanhove et al., 2011, 2012) and congrids (Kobashi et al., 2004; Ivany et al., 2000, 2003; De Man et al., 2004; Vanhove et al., 2011, 2012). Because these groups are thought to be benthic and nonmigratory throughout their lifespan, they are expected to record the seasonality within a single environment.

Nevertheless, due to taxon-specific differences in mode of life, preferred habitat and season of growth, different types of skeletal archives in a single death assemblage may not uniformly record similar ambient conditions. Moreover, many
sclerochronological records are based on fossils from shallow marine settings, which can be characterized by local, lateral, and temporal variations in water composition and stability, caused by riverine freshwater influx, longshore and tidal currents, upwelling, evaporation, and temperature extremes, exacerbating the potential differences between various local microhabitats or ecologies. All of this could lead to differentiations in isotopic variability between different taxa. On top of this, due to temporal averaging in sedimentary records, specimens from the same death assemblage may have experienced different
environmental conditions during life. Therefore, to allow accurate reconstructions of seasonality using sclerochronological records, it is essential to tightly constrain the paleoecology and –environment of these carbonate-secreting organisms.

In order to assess variation in sclerochronological stable isotope profiles in commonly used archives in the Paleogene, this study focuses on seasonal variation within and variability among incremental $\delta^{18}$O and $\delta^{13}$C profiles of a turritelline gastropod species (*Haustator solanderi*), two species of carditid bivalves (*Venericor planicosta lerichei* and *Cyclocardia*
*(Arcturellina) sulcata aizyensis*) and otoliths from two species of demersal non-migratory fish, the congrid *Paraconger sauvagei* and the ophidiid '*Neobythites*' *subregularis* (Fig. 1). All fossils are derived from two successive upper Ypresian shallow marine coquina beds in the inner neritic Aalter Sand Formation (Zenne Group), outcropping in the Belgian part of the southern North Sea Basin (sNSB). The lower coquina bed contains abundant large *Venericor* bivalve shells, while the upper one contains abundant *Haustator* gastropods. At Aalter, the coquina beds, which provided the studied material, are separated
by an omission surface, implying that the *Venericor* bed belongs to the top of Unit A2 and the *Haustator* bed to the lower



middle part of Unit A3, as defined in Steurbaut & Nolf (2021). Each coquina is likely deposited within hundreds, to possibly thousands, of years, while the succession containing both coquinas, encompassing the top of Unit A2 and the lower part of Unit A3, is probably deposited within a time span of several tens of thousands of years. Hence, all fossils are derived from a narrow, stratigraphically well-constrained interval, corresponding in time to the latest Ypresian, at ~49.2 Ma (Steurbaut and

Nolf, 2021), just after the early Eocene climatic optimum (EECO; Zachos et al., 2008; Evans et al., 2018; Speijer et al., 2020). Given this narrow stratigraphic interval, it can be assumed that similar climatological conditions and environmental settings prevailed during deposition of both coquina beds. All fossils from these coquina beds are therefore expected to record similar climatological and environmental conditions.

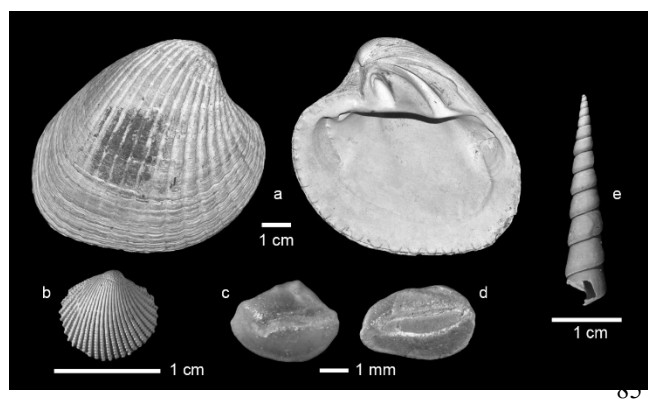

**Figure 1: Representative specimens of taxa from the Aalter Sand Fm. used in this study: a)** *Venericor planicosta lerichei*, **specimen B13D; b)** *Cyclocardia* (*Arcturellina*) *sulcata aizyensis*, **specimen B9B; c)** *Paraconger sauvagei*, **specimen O205B; d) '***Neobythites***' *subregularis*, **specimen O207B; e)** *Haustator solanderi*, **specimen B11H.**

We explore environmental, biological, ecological and taphonomical causes for differences in amplitude and mean of intra-annual isotope variability among specimens and species. Our findings lead to a better understanding of isotopic variability between different types of skeletal archives in a death assemblage from a shallow marine setting, and of the constraints on
their value as proxy archives of seasonality.

## 2 Geological context

The Aalter Sand Formation consists of a grey green, glauconiferous, clayey fine sand, locally with fine sandy clay layers and a few thin poorly cemented sandstone layers, which passes into very fossiliferous, fine grey sand at the top (Steurbaut and Nolf, 1989). It crops out locally in northwestern Belgium in the Gent-Aalter-Oedelem area and in a series of small hills 60 km
to the southwest along the French-Belgian border (Scherpenberg, Cassel, Mont-des-Récollets), and occurs in the subsurface in the southwestern part of the Netherlands and offshore on the Belgian continental shelf (Le Bot et al., 2003; Steurbaut and Nolf, 2021). In the mid-1800's, first descriptions of fossils in the Aalter Sand Formation appeared in field trip reports, emphasizing the abundance of turritellines and large *'Cardita (Venericardia) planicosta'* shells near the Aalter train station in



Belgium (e.g. Nyst and Mourlon, 1871). Today, parts of the Aalter Sand Fm. are occasionally exposed during construction

works in the Aalter region.

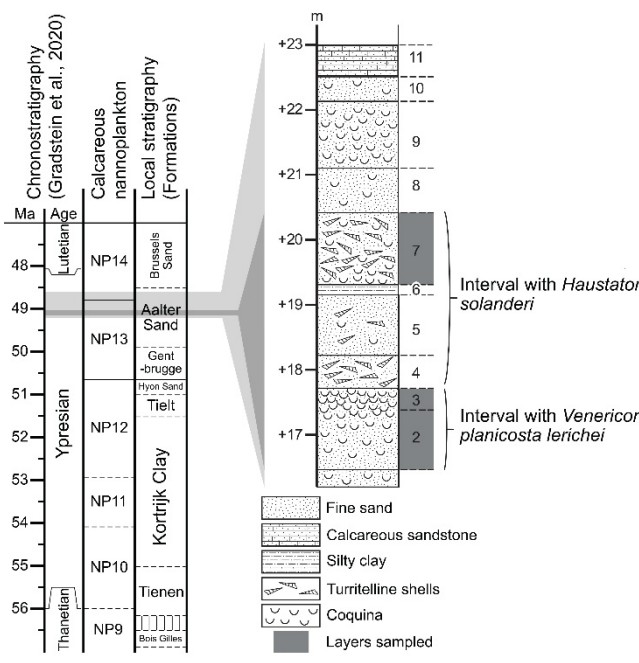

**Figure 2: Part of the stratotype of the Aalter Sand Fm. at Aalter, Belgium (modified after Steurbaut & Nolf, 1989). The layers from which specimens used in this study were taken correspond to layers 2-3 and 7 of the stratotype, correlated to the chronostratigraphic timescale.**

This study is limited to two coquinas in the Oedelem Member of the Aalter Sand Formation (Steurbaut and Nolf, 1989; King, 2016), cropping out in the Aalter stratotype area. The studied coquinas correspond to the interval with

'*Megacardita planicosta lerichei*' [now *Venericor planicosta lerichei* (GLIBERT and VAN DE POEL, 1970)] and the interval with '*Turritella solanderi*' [now *Haustator solanderi* (MAYER-EYMAR, 1877)], layers 2-3 and layers 4-7 in Steurbaut & Nolf (1989) (Fig. 2), respectively. Henceforth, the two coquinas are referred to as the 'venericard' and 'turritelline' layers. They are both composed of olive-green fine-to-medium grained and locally coarse sands with glauconite and abundant mollusks. The shells of *V. planicosta lerichei* occur mainly disarticulated, but occasionally as double-valved specimens,

suggesting quick post-mortem burial (Steurbaut and Nolf, 1989). In the turritelline-bearing unit, an estimated 90% of the macrofossils belong to the species *H. solanderi*. This unit convincingly meets the criteria for a turritelline-dominated assemblage (TDA), as *H. solanderi* comprises more than 20% of the estimated biogenic carbonate, and is more than twice as abundant as any other macroscopic species in the assemblage (Allmon, 2007). Other constituents in the Aalter Sand Fm. fauna include benthic foraminifera, bryozoans, irregular echinoids, solitary corals, serpulid worm tubes, other gastropods and

bivalves, cephalopod fragments, teleost fish otoliths and bone fragments, and chondrichthyan teeth (Nyst and Mourlon, 1871; Kaasschieter, 1961; Nolf, 1972a; Steurbaut and Nolf, 1989). Calcareous nannofossil assemblages correspond to the upper part of zone NP13 (Steurbaut and Nolf, 1989, 2021). The first occurrences of *Discoaster* cf. *sublodoensis* (a few corroded *D. sublodoensis*-like specimens) and *Discoaster praebifax* are situated close to the base of the turritelline-bearing interval





(Steurbaut and Nolf, 1989; King, 2016), while the last occurrences of *Nannoturba jolotteana* and *N. spinosa* occur above the venericard layer (Steurbaut and Nolf, 2021), placing the studied interval in top NP13b-base NP13c of Steurbaut & Nolf (2021). The first occurrence of *Discoaster sublodoensis* s.s., part of the major calcareous nannofossil turnover around the Ypresian-Lutetian boundary (Steurbaut and Nolf, 2021), occurs above the studied interval. The age of the coquinas is therefore latest Ypresian, around 49.2 Ma, with respect to the Paleogene time scale (Speijer et al., 2020).

## 3 Paleoenvironment and paleogeography

### 3.1 Paleoenvironment

The general environmental model for the late Ypresian – early Lutetian in the Belgian part of the sNSB is an advancing and retreating deltaic system (Jacobs et al., 1991; Jacobs and Sevens, 1993; Gibbard and Lewin, 2016). The combination of sediment supply from this delta and sea-level variations led to an array of highly diverse facies representing inner neritic, sublittoral, intertidal, barrier, lagoonal and estuarine deposits (Jacobs et al., 1991). Also the Aalter Sand Formation shows a

large facies diversity, both vertically and laterally (Jacobs and Sevens, 1993), reflecting these varying sedimentary environments. Nonetheless, the presence of glauconite, characteristic sedimentology and diagnostic fossil content, including calcareous nannoplankton assemblages indicative of a shallow water conditions (Steurbaut and Nolf, 1989; Jacobs and De Batist, 1996), unambiguously establish deposition in a very shallow marine, littoral or sublittoral environment. The overall otolith assemblage occurring in the Aalter Sand Fm. (Nolf, 1972a) is in agreement with this. While modern *Paraconger* has

been recorded from 10-75 m (Kanazawa, 1961) and ophidiids occur from shelf to abyssal depths (FishBase, 2023), recent relatives of the genera *Orthopristis*, *Platycephalus* and *Apogon*, all present in the Aalter assemblage (Nolf, 1972a), occur in very proximal habitats (FishBase, 2023). The nannofossils (Steurbaut and Nolf, 1989), otolith assemblage (Nolf, 1972a), mollusk assemblages (Nyst and Mourlon, 1871; Glibert, 1985) and presence of solitary corals and cephalopod fragments (Steurbaut and Nolf, 1989), together with storm-induced winnowing and occasionally subtle channeling in the venericard

coquina, all suggest a shallow, inner neritic depositional setting, above fair-weather base (Jacobs and Geets, 1977), probably less than 30 m deep, with fully marine, normal salinity conditions.

Jacobs and Sevens (1993) interpreted the sequence of the Oedelem Member of the Aalter Sand Fm. in boreholes offshore the current Belgian coast as a submerged coastal barrier, followed by storm deposits in a lagoonal environment, as evidenced by partially incising fine sands with centimeter thick clay layers and flaser lamination. The accumulations of

venericard and turritelline shells were interpreted to reflect either storm or chenier (see Cangzi and Walker, 1989) deposits. Nevertheless, in the stratotype area, the Aalter Sand Formation contains less clay, and flaser bedding was not observed (Jacobs and Geets, 1977; Steurbaut and Nolf, 1989), with granulometry indicating non-uniform low-turbulence suspension (Jacobs and Geets, 1977). These results are more consistent with a subtidal channel system, potentially with lagoons or mudflats relatively nearby. The coquina beds likely represent concentrations of shells on the bottom of these channels, resulting from

winnowing and/or storms (Steurbaut and Nolf, 1989; Jacobs, 2015).



## 3.2 Paleoceanography

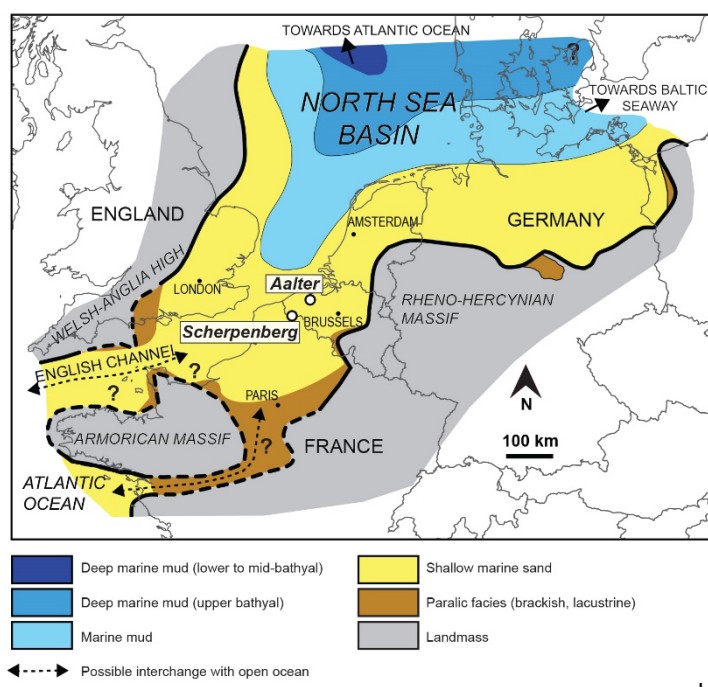

**Figure 3: Paleogeographic reconstruction of the southern North Sea Basin area around the latest Ypresian to early Lutetian. The basin had two connections with the Atlantic Ocean, one to the north and one to the southwest. There are two options for a southwestern gateway, one via the English Channel and the other south of the Armorican Massif ('Loire Seaway'). Paleogeography based on and modified from (Murray, 1992; King, 2006; Gély, 2008; Knox et al., 2010; Huyghe et al., 2012; Steurbaut et al., 2016).**


The paleolatitude of Aalter is ~41°N (van Hinsbergen et al. 2015). During the Eocene, the sNSB area shared similarities with today's configuration as a partly enclosed, shallow siliciclastic shelf environment. However, it differed in terms of the number and position of connections to the open ocean (Fig. 3). In the late Ypresian to early Lutetian, the sNSB probably opened northward to the Atlantic Ocean and Arctic waters via the Viking Graben and Norwegian Seaway (Knox et al., 2010; Gibbard

and Lewin, 2016). In addition, microfossil evidence and the presence of several fish taxa primarily associated with the Indo-West-Pacific region, including the genus *Platycephalus*, provide support for an early to mid-Ypresian connection between the sNSB and the Tethys region to the southeast (Nolf, 1972b; Steurbaut and Nolf, 1990; Steurbaut, 2011; Knox et al., 2010; King, 2016). Also the Neobythitinae, a subfamily of cusk eels, are thought to have entered the western Atlantic via the eastern Pacific (Nielsen, 1999; Nielsen et al., 1999), indicating that their Eocene arrival in the North Sea Basin must have been through an

eastward or southward connection with the Tethys area, via the Baltic Seaway or the precursor of the English Channel (Knox et al., 2010). Evaluating whether this eastward connection maintained during the late Ypresian proves challenging, as the discrepancies in calcareous nannofossil content between the North Sea Basin and the Peritethys during that time could either indicate a severed connection or the emergence of distinct paleoenvironmental settings (Steurbaut, 2011; King, 2016).

From the late Ypresian onwards, the southward connection with the Atlantic Ocean was impacted by tectonically

controlled uplift related to the renewal of Africa-Europe convergence. This uplift, indicated by the erosive bases of the Egem Mb. and Brussels Fm. and the presence of paleoseismites in these units, caused restricted connection between the Belgian and Paris Basins (Vandenberghe et al., 2004). King (2006) however, in support of an 'open' configuration for the sNSB, lists



several lines of evidence for continued interchange with the Atlantic, most importantly the presence of warm-water nummulites in the sNSB and the fact that no physical barrier is required to explain differences in facies between the London and Hampshire Basins. Two possible options for a southward connection with the Atlantic are through the English Channel or, if this channel would have been blocked by uplift of the Start-Cotentin Swell, via the Loire Seaway (Gély, 2008; Knox et al., 2010; Huyghe et al., 2012). Nevertheless, during the early Eocene sNSB region likely became increasingly enclosed, possibly influencing the local sea water isotopic composition ($\delta^{18}O_{sw}$).

## 4 Methods

### 4.1 Specimen collection

Specimens were retrieved from field sampling and from the collections of the Belgian Institute of Natural Sciences (INS; see Table 1 and Fig. 1). In 2010, about 3 m of the Aalter Sand Fm. below layer 11 (Steurbaut and Nolf, 1989) was exposed in a construction pit located in the village center (Hagepreekstraat, Aalter, Appendix A). The lower part of the exposure contained abundant *H. solanderi*, and is therefore thought to correspond to layer 7 in the Aalter Sand Fm. stratotype description Steurbaut and Nolf (1989). Most of the specimens studied here were sorted from a 30 kg sediment sample residue of this turritelline coquina, including six *H. solanderi* specimens, one specimen each of the otoliths from *Paraconger sauvagei* (PRIEM, 1906) and '*Neobythites*' *subregularis* (SCHUBERT, 1916), and nine small carditids belonging to *Cyclocardia (Arcturellina) sulcata aizyensis* (DESHAYES, 1858). Taxonomy of otoliths is according to Lin et al. (2017). When otolith genera are written with quotation marks it concerns a morphologically delineated species that cannot be assigned to a specific genus, but only to a higher rank, in this case, the subfamily Neobythitinae. However, because of the demand for adopting a strictly binomial nomenclature, '*Neobythites*' is now preferred above "genus *Neobythitinorum*", a term formerly used in Vanhove et al. (2011, 2012). Additional specimens of *P. sauvagei* and '*N.*' *subregularis*, one of each (O205B and 0208B), were selected from the INS fish otolith collection, collected around 1970 by dr. Dirk Nolf in the turritelline coquina, a few hundred meters west of our sample locality (Nolf, 1972b). From the venericard layer, three large *Venericor planicosta lerichei* (GLIBERT and VAN DE POEL, 1970) specimens were selected from the INS invertebrate collection. Previously published results from two ophidiid specimens (Vanhove et al., 2011) from the Aalter Sand Fm. at Scherpenberg (O2B; O2D; Fig. 3) are included in the dataset and the results section, to facilitate data comparison and to provide a more complete view of the lateral isotopic variability in the Aalter Sand Fm.. The Scherpenberg locality is located 56 km SW of Aalter and precise correlation with the stratotype in the Aalter locality has not been attempted (Nolf, 1972a).



**Table 1: Collection and sample metadata of the specimens used in this study, retrieved from field sampling and museum collections.**

| ID* | Species | Family | Locality | Collection | Lithostratigraphy | Sample type |
|---|---|---|---|---|---|---|
| **Otoliths:** | | | | | | |
| O205B | *Paraconger sauvagei* | Congridae | Aalter, B., RMS*, Stationstraat | RBINS** Otolith coll. | level w. *H. solanderi* | increments |
| O206A | *Paraconger sauvagei* | Congridae | Aalter, B., Hagepreekstraat | new sampling | level w. *H. solanderi* | increments |
| O207B | *'Neobythites' subregularis* | Ophidiidae | Aalter, B., Hagepreekstraat | new sampling | level w. *H. solanderi* | increments |
| O208B | *'Neobythites' subregularis* | Ophidiidae | Aalter, B., RMS*, Stationstraat | RBINS Otolith coll. | level w. *H. solanderi* | increments |
| O2B*** | *'Neobythites' subregularis* | Ophidiidae | Scherpenberg, B. | RBINS Otolith coll. | - | increments |
| O2D*** | *'Neobythites' subregularis* | Ophidiidae | Scherpenberg, B. | RBINS Otolith coll. | - | increments |
| O2E | *'Neobythites' subregularis* | Ophidiidae | Scherpenberg, B. | RBINS Otolith coll. | - | XRD only |
| **Gastropods:** | | | | | | |
| B11A, C, E, F-H | *Haustator solanderi* | Turritellinae | Aalter, B., Hagepreekstraat | new sampling | level w. *H. solanderi* | increments |
| **Bivalves:** | | | | | | |
| B13B | *Venericor planicosta lerichei* | Carditidae | "Aeltre, abri face à la maison communale" | RBINS Invert-13660-0002 | level w. *V. planicosta* | bulk lines |
| B13D | *Venericor planicosta lerichei* | Carditidae | "Aeltre, abri face à la maison communale" | RBINS Invert-13660-0004 | level w. *V. planicosta* | increments |
| B14E | *Venericor planicosta lerichei* | Carditidae | "Aeltre" | RBINS Invert-03280-0001 | level w. *V. planicosta* | increments |
| B9A-F, H-J | *Cyclocardia (Arcturellina) sulcata aizyensis* | Carditidae | Aalter, B., Hagepreekstraat | new sampling | level w. *H. solanderi* | bulk |

*RMS = Rijksmiddelbare School
**RBINS = Royal Belgian Institute of Natural Sciences
***from Vanhove et al. (2011)


## 4.2 Assessing preservation

The sediments of the Aalter Sand Fm. are unlithified and have not been buried deeply since deposition. Diagenetic alteration of primary aragonite is therefore not to be expected. Indeed, previous studies have already demonstrated the excellent

preservation of otoliths from the Ypresian deposits in Belgium, including from the Aalter Sand Fm. at Scherpenberg, using scanning electron microscopy (SEM), cold cathode luminescence microscopy and X-ray diffraction (XRD) analyses (e.g. Vanhove et al., 2011). In order to confirm the preservation state of the specimens studied here, we performed optical microscopy on all studied specimens, and XRD analyses on powdered fragments from *H. solanderi* and *V. planicosta lerichei* specimens from Aalter and a '*N.' subregularis* specimen from Scherpenberg, by means of a Philips PW 1830 Generator (CuKα,

30 mA, 45 Kv, 15-55° detection), at the Department of Earth and Environmental Sciences (KU Leuven), using Topas Academic V4 software for comparison with standard patterns. In addition, the same specimens of *H. solanderi* and *V. planicosta lerichei* from Aalter were also investigated using SEM (Jeol JSM-6400).

## 4.3 Microsampling and isotope analysis

Before microsampling, all specimens were ultrasonically rinsed in deionized water. Otoliths were embedded with an automated

mounting press and polished manually on wetted sand paper until slightly above the sagittal plane such that a maximum surface and depth is available for microsampling (see Vanhove et al., 2011).



*Venericor planicosta lerichei* bivalves were embedded in epoxy resin manually, cut with a wetted slow-speed saw (Syracuse University, NY) and finely polished to reveal growth features. Otoliths and *V. planicosta lerichei* bivalves were sampled with a Merchantek (currently ESI) micromilling device with 0.1 mm (Brasseler®) or 0.3 mm (Dremel®) drill bits

(Vrije Universiteit Brussel, VUB). Incremental samples were drilled parallel to growth increments, and represent continuous series. In otoliths, approximately ¾ of the sagittal plane was sampled from nucleus to edge. In *V. planicosta lerichei*, three major increments, presumably corresponding to years, were sampled in the umbonal region. We calculate specimen means and ranges from data encompassing at least three annual cycles. Sampling a limited number of years from multiple specimens is more representative than sampling many years in only one individual, and accounts for taphonomic redistribution and

averaging of the depositional time window (Goodwin et al., 2003). In specimen B13D, years 9, 10 and 11 were selected out of an estimated total number of 13+ years. Specimen B14E lived for 9 years and years 6, 7 and 8 were selected. Bulk samples were taken in specimen B13B by drilling approximately linear paths perpendicular to all growth lines, in the middle part of the ventral region. As *Cyclocardia (A.) sulcata aizyensis* specimens are smaller than 1 cm in diameter, only bulk samples could be generated, by taking a subsample of powder resulting from crushing and homogenizing the entire shell.

Haustator solanderi specimens were sampled along the whorls with a Dremel® hand-held drill progressively at constant angles of mostly 180° or 90° along the outer shell from apex to aperture. Specimens B11F and B11H were sampled with a resolution of four samples per whorl, while only two samples per whorl were drilled in specimens B11A, B11C, B11E and B11G. Specimen B11A could not be fully sampled because the first series of whorls broke off during microdrilling; two other specimens had incomplete apexes (B11C, B11G), impeding a sampling of the first whorls.

The weight of the microsampled aragonite powders varied between 40 and 80 µg and was analyzed in the stable isotope laboratory of the VUB with a ThermoFinnigan Kiel III coupled to a DeltaPlusXL mass spectrometer. Average precision based on NBS19 replicates is 0.02 ‰ for carbon and 0.05 ‰ for oxygen. A number of replicate samples of the *C. (A.) sulcata aizyensis* specimens were run in the SIL of the University of Michigan, Ann Arbor, MI, with a Kiel IV coupled to a ThermoFinnigan MAT 253 mass spectrometer. Results are reported in ‰ VPDB, and values for $\delta^{18}O_w$ in ‰ VSMOW. Raw

results of all analyses are listed in Supplement S1.

For statistical comparison between results from different taxa the unpaired t-test for equal means was used, unless the F-test indicated unequal variances, in which case the Welsh test was used. Significance level α was 0.05 in all cases.

The ShellChron 0.4.0 model (De Winter, 2022), based on the approach of Judd et al. (2018), was used in R (R version 4.3.1, Rstudio version 2023.06.0 build 421) to transfer $\delta^{18}O_c$ data to the time domain (Julian days), using a combination of

growth rate and temperature sinusoid models on the isotope records while applying a sliding window approach. For this, the model makes the assumption that growth and temperature follow quasi-sinusoidal patterns (De Winter, 2022). Since variability in $\delta^{18}O_w$ is comparatively limited in most fully marine environments (Rohling, 2013), the $\delta^{18}O_c$ record preserved in the accretionary shells of extra-tropical mollusks can offer a good approximation of seasonal temperature fluctuations so long as season of growth is taken into account (Judd et al., 2018; Ivany and Judd, 2022; De Winter, 2022).



## 5 Results

### 5.1 Preservation

Optical microscopic examination reveals that even though otoliths from the Aalter Sand Formation do show slight signs of abrasion, the fossils from this formation are generally well-preserved. The *V. planicosta lerichei* shells show clear macroscopic growth increments. XRD of powdered fragments from *H. solanderi* and *V. planicosta lerichei* specimens from Aalter and a '*N.*' *subregularis* specimen from Scherpenberg, indicate that they are composed entirely of aragonite (Appendix B). SEM images of the same *H. solanderi* and *V. planicosta lerichei* specimens from Aalter reveal ultrastructures similar to those of modern specimens (Appendix B). It is therefore unlikely that the aragonite shells and otoliths from the Aalter Sand Fm. are significantly affected by dissolution and/or chemical alteration.

### 5.2 δ18O results

Distinct intra-annual variation is observed in all $\delta^{18}O$ profiles, with a total range over all specimens of 4.7 ‰, from -5.5 to -0.8 ‰ (Figs. 4-7). The results of the ShellChron 0.4.0 model reveal seasonal fluctuations of oxygen isotope levels throughout the year per species, while also depicting the variation between their reconstructions (Fig. 8).

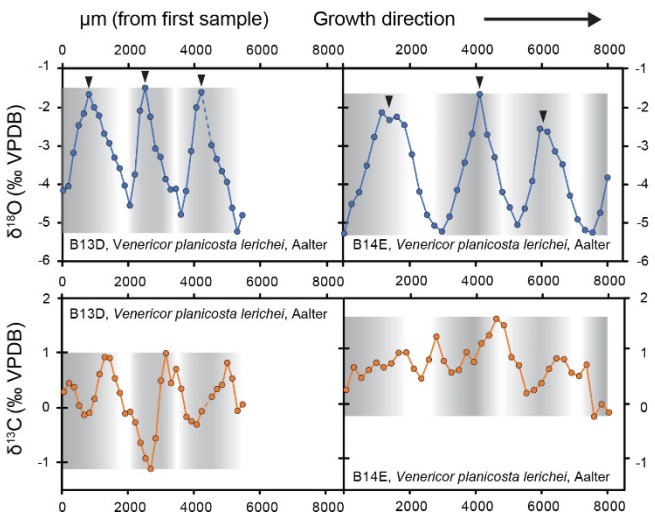

**Figure 4: δ13C and δ18Oc results of two serially sampled *Venericor planicosta lerichei* specimens (B13D, B14E) from the Aalter Sand Fm., Belgium. Growth increments are shown as gradient bars behind the isotope profiles. Year markers (black triangles) indicate interpreted winter peaks.**

315



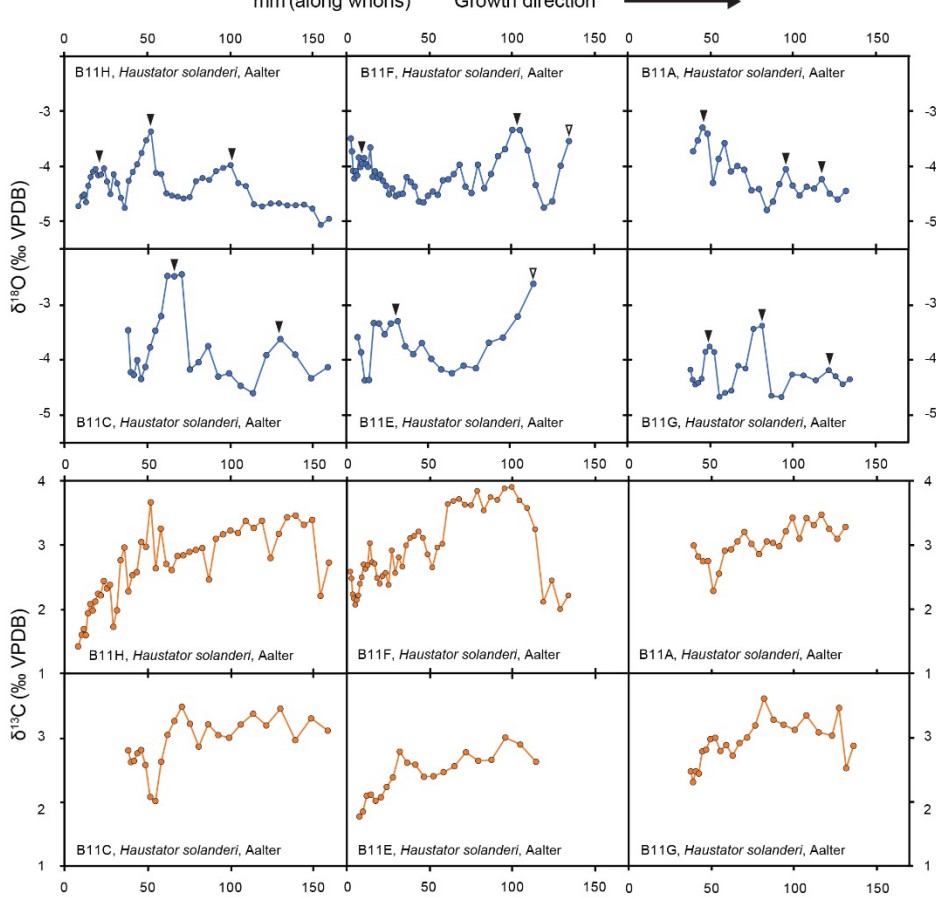

**Figure 5: δ13C and δ18Oc results of six serially sampled *Haustator solanderi* specimens from the Aalter Sand Fm., Belgium. Year markers (black triangles) indicate interpreted winter peaks, open triangles indicate possible winter peaks.**



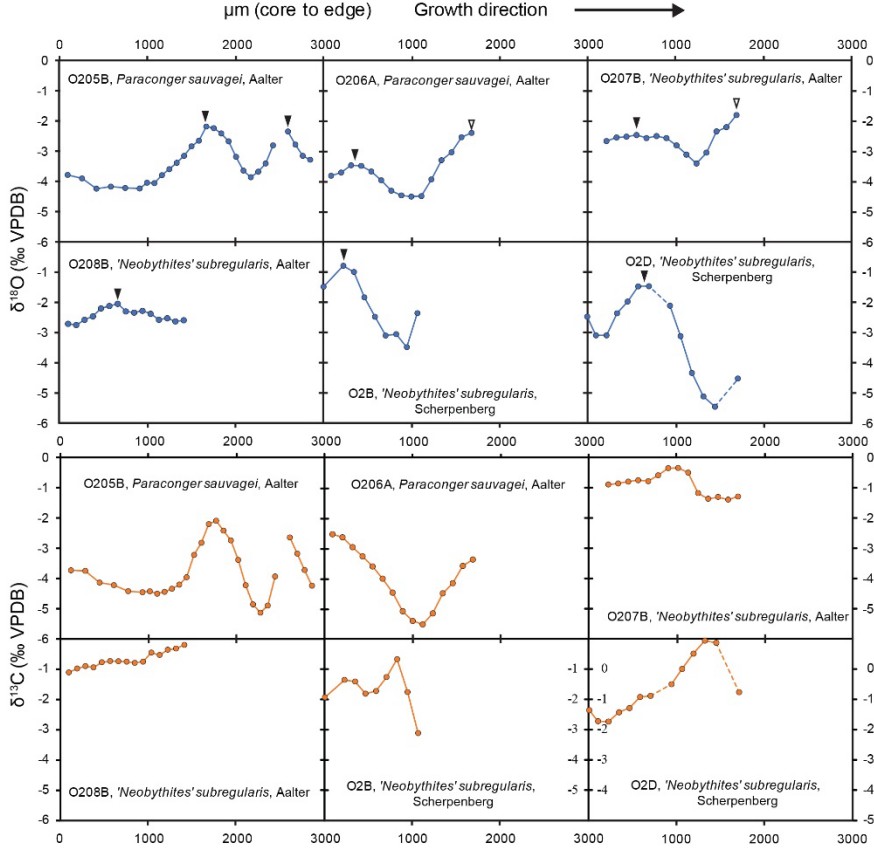

**Figure 6: δ13C and δ18Oc results of serially sampled *Paraconger sauvagei* (O205B, O206A) and '*Neobythibites*' *subregularis* (O207B, O208B, 02B, O2D) specimens from the Aalter Sand Fm. at Aalter and Scherpenberg, Belgium. Year markers (black triangles) indicate interpreted winter peaks, open triangles indicate possible winter peaks**



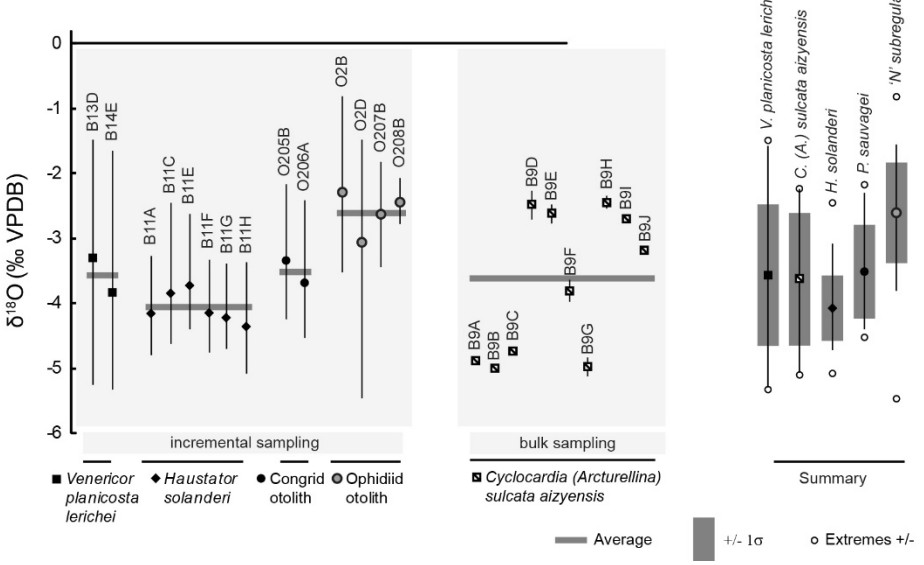

**Figure 7: δ18Oc ranges for all sampled specimens of the Aalter Sand Fm. at Aalter and Scherpenberg, Belgium. Average and maximum specimen range of individual specimens are shown on the left, and summaries for each taxon (average, standard deviation, average minima and maxima, extremes) on the right.**

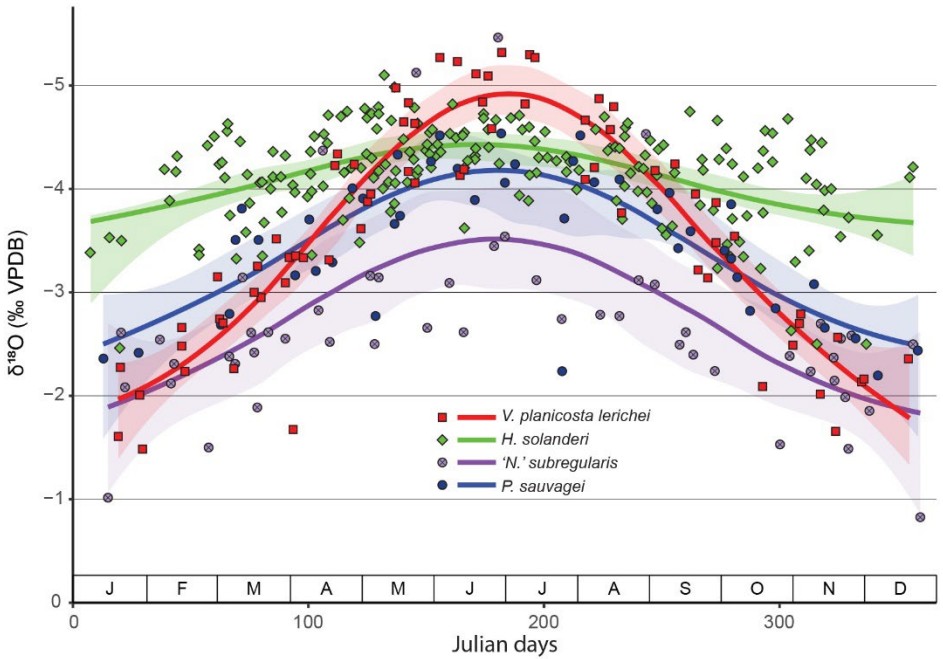

**Figure 8: Mean annual cycles for each taxon generated by ShellChron 0.4.0 using δ18Oc data for *H. solanderi*, *V. planicosta*, *P. sauvagei* and '*N.*' *subregularis*. Lines represent a sinusoidal regression curve generated based on the data of each species. Ribbons represent the 95% confidence intervals of the different species. Letters J-D represent months.**

Stable oxygen isotope profiles of the large carditid bivalve *V. planicosta lerichei* reveal prominent, and relatively consistent, cyclical variation. The shape of the profile is sinusoidal or saw-toothed. Values in both specimens range from about





-1.5 to -5.3 ‰, spanning the largest range in intra-annual values observed for all taxa. The average of the 5 percent lowest

values is -5.28 ‰, the 5 percent higher values is -1.60 ‰. The species mean for *V. planicosta lerichei* is -3.6 ‰. Results of

whole shell analyses of the small carditid bivalve *C. (A.) sulcata aizyensis* vary between -2.4 and -5 ‰ and show a species

average of -3.3 ‰ (see Supplement S1). The average of the 5 percent lowest values is -5.05 ‰, the 5 percent higher values is

-2.34 ‰. While this range of values falls within the range of *V. planicosta lerichei*, it is striking that the different *C. (A.) sulcata*

*aizyensis* specimens have such large differences in average values. The oxygen isotope values of subsamples from the

turritelline gastropod *H. solanderi* range between -2.5 and -5.1 ‰, with a species average of -4.1 ‰. The average of the 5

percent lowest values is -4.80 ‰, the 5 percent higher values is -2.97 ‰. The average intra-annual range of this species is 1.7

‰, smaller than the other studied taxa. Nevertheless, the amplitude of variation varies considerably between different *H.*

*solanderi* shells, ranging from 2.2 ‰ in B11C to just 1.3 ‰ in B11G. The profiles of *H. solanderi* are more variable and less

consistent in terms of shape and cyclicity compared to the bivalves from this location. Some specimens (B11F, B11C, B11E)

show a cuspate pattern, with sharp peaks and broad troughs. In otoliths from Aalter, cyclicity is most pronounced in the two

congrid specimens, with values ranging between -2.2 and -4.5 ‰, and a species average of -3.5 ‰. The average of the 5 percent

lowest values is -4.51 ‰, the 5 percent higher values is -2.19 ‰. The oxygen isotope profiles are again relatively cuspate. The

two ophidiid specimens from Aalter are characterized by values in between -1.8 and -3.4 ‰ and a species average of -2.5 ‰,

a smaller range and more positive values in comparison with the congrid specimens. The average of the 5 percent lowest values

is -3.43 ‰, the 5 percent higher values is -1.82 ‰. Two ophidiid specimens from the Scherpenberg locality (Vanhove et al.,

2011) have an intra-annual range between -0.8 and -5.5 ‰, larger than the Aalter ophidiid and congrid specimens, and show

a species average of -2.8 ‰, lower than the ophidiids from Aalter. The taxon means of the two species of carditid bivalves (*V.*

*planicosta lerichei* and *C. (Arcturellina) sulcata aizyensis*) and the otolith species *Paraconger sauvagei* are statistically the

same (Welsh test: p<0.01), the means of the other groups are statistically different.

## 5.2 δ¹³C results

In general, intra-annual variation in δ¹³C is less consistent than in δ¹⁸O (Figs 4-6, 9; Supplement S1). The total range

across all specimens extends from -5.6 to +3.9 ‰, spanning about 9.5 ‰. Within each taxon, the range in δ13C is much lower,

about 3 ‰ or less. Specimens of *H. solanderi* have the highest values (average of +2.8 ‰), followed by *C. (A.) sulcata* (average

of +1.9 ‰), *V. planicosta lerichei* (average of +0.5 ‰) and the ophidiid otoliths (average of -1.0 ‰), with the lowest values

observed in the congrid otoliths (average of -3.8 ‰). T-tests, or in case of unequal variance Welsh tests, indicate that all taxon

means are statistically different. Intra-annual range of δ13C is generally smallest in ophidiid otoliths from Aalter (average

range of 1.0 ‰), followed by *H. solanderi* (average range of 1.6 ‰), *V. planicosta lerichei* (average range of 2.0 ‰) and

congrids (average range of 2.1 ‰), whereas ophidiids from Scherpenberg show the largest average range, of 2.6 ‰. While

seasonality is apparent in the otolith δ¹³C profiles, in particular in the congrids, intra-annual δ¹³C variability in *H. solanderi*

and *V. planicosta lerichei* does not exhibit a distinct seasonal signal. Instead, in line with previous studies on Eocene mollusks

(e.g. De Winter et al., 2020; Clark et al., 2022), the δ¹³C variability in mollusks (bivalves and gastropods) from the Aalter Sand



Fm. displays significant variability within seasons and lacks a consistent pattern among the specimens. This suggests that these δ13C fluctuations are not exclusively related to annual or seasonal changes in the δ13C of dissolved inorganic carbon, but
instead are more likely heavily influenced by vital effects (McConnaughey and Gillikin, 2008). Notably in the turritellines, all δ13C patterns show a marked increase in δ13C over their life span (Fig. 5), similar to other Eocene turritelline records from northwest Europe (Andreasson and Schmitz, 1996, 1998, 2000).

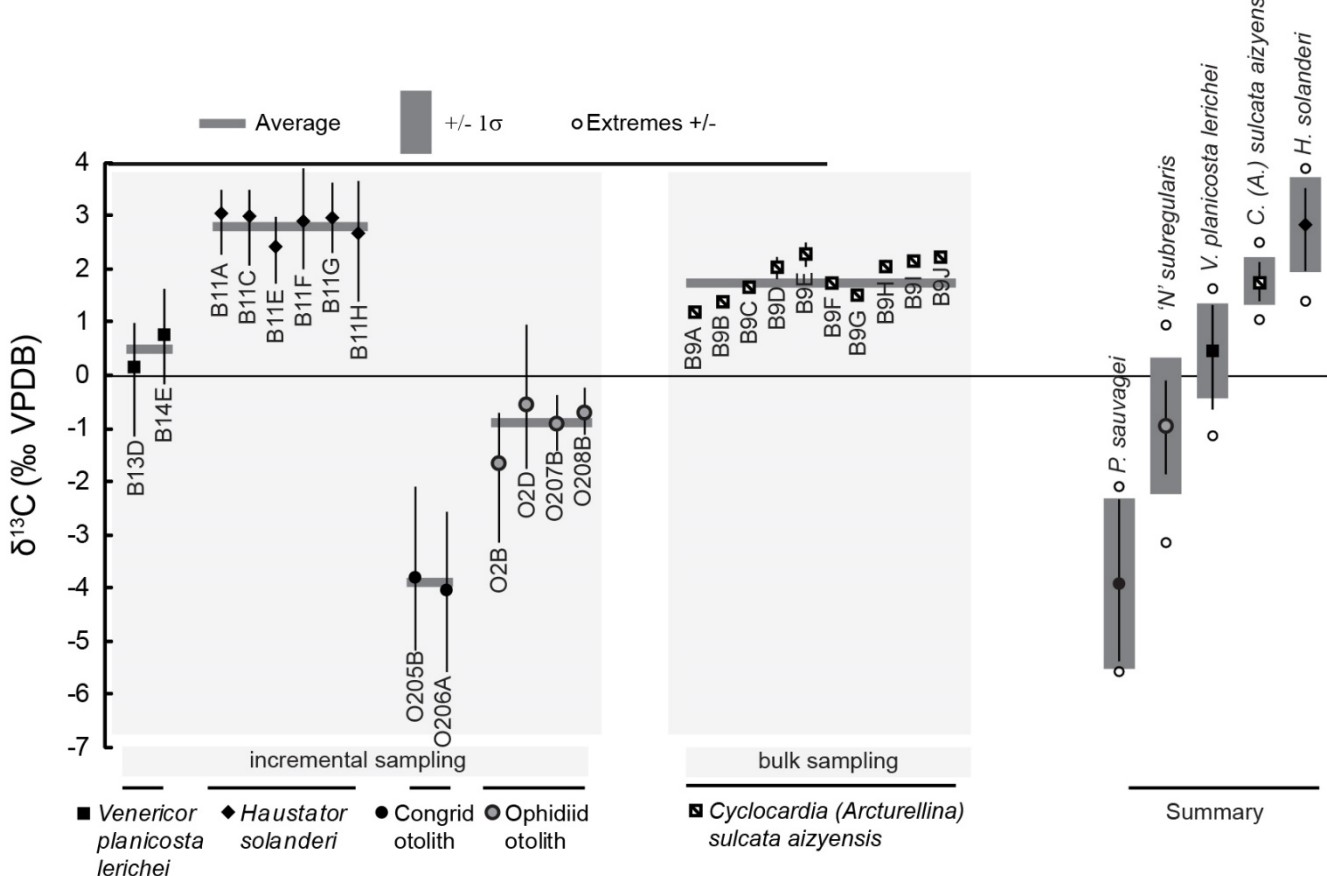

**Figure 9: δ13C ranges for all sampled specimens of the Aalter Sand Fm. in Aalter and Scherpenberg, Belgium. Average and maximum specimen range of individual specimens are shown on the left, and summaries for each taxon (average, standard deviation, average minima and maxima, extremes) on the right.**



## 6 Discussion

### 6.1 General observations

While all studied fossils from the Aalter locality should hypothetically record the same climatological conditions and environmental setting on average, their oxygen isotopic profiles show large differences, with species averages ranging from -4.1 ‰ in the turritelline gastropod *H. solanderi* to -2.5 ‰ in the ophidiid '*N.*' *subregularis*. Minimum values are comparable across *V. planicosta*, *H. solanderi*, and *P. sauvagei* and generally fall between -5.5 ‰ and -4.5 ‰, the maximum values in these taxa range from -3.1 ‰ in *H. solanderi*,  to -1.6 ‰ in  *V. planicosta*, resulting in average annual ranges as large as 3.7 ‰ in *V. planicosta* to just 1.7 ‰ in *H. solanderi*. The '*N.*' *subregularis* specimens from Aalter deviate in both minimum and maximum values, -3.4 ‰ and -1.8 ‰, respectively. The range in average specimen values (-2.3 to -4.4 ‰), statistical difference between most taxon means, and variability in specimen ranges could mean that organisms recorded different conditions during their lifetimes, or over the period of time covered by the death assemblage in which they occur. Either way, organisms preserved in the Aalter coquina beds precipitated carbonate in waters of varying isotopic composition or temperature, or a combination of both. In this and the subsequent sections, we explore the ecological, paleoenvironmental and taphonomical factors that likely contribute to the observed variability in intra-annual and average $\delta^{18}O$ recorded by the different taxa from the Aalter coquinas.

### 6.2 Taphonomy

Both coquinas are assumed to have been deposited within a relatively short time interval, probably within hundreds to thousands of years each. The overall good preservation of the fossil assemblage is consistent with relatively rapid deposition. Nevertheless, because the coquina beds were formed as concentrations of shells, likely on the bottom of subtidal channels by current or storm-induced winnowing, they likely represent an amalgamation of material transported from various local microhabitats or nearby environments over some number of years. If these settings experience spatio-temporal differences in the mean and variation of temperature and/or salinity, and/or if fossils are variably derived from different settings, this could contribute to the variability seen among our samples.  The complex depositional environment of the Oedelem Member, with subtidal channels and potential lagoons or mudflats nearby, comprises a variety of local habitats and is subject to storm wave action, which could well be responsible for differences in average and intra-annual range in $\delta^{18}O$ between individual specimens.

When considering the different studied fossil groups, not all are likely to have been transported to the same degree. In general, the good preservation of turritellines and carditids in the Aalter Sand Fm. coquinas argues in favor of limited reworking or transport. *Venericor planicosta lerichei* shells are heavy and large (6-10 cm length) in comparison with the turritelline shells (3.5 cm height), small carditids (*Cyclocardia*; <1 cm length) and otoliths (3-5 mm length), suggesting a generally lower post-mortem transportability. Sessa et al. (2012) found higher isotopic variability amongst small venericards in the early Eocene of the US Gulf Coast in comparison with large co-occurring specimens, and they suggested transportation



of smaller shells as well. Notably, towards the top of the venericard coquina in Aalter, articulated valves of *V. planicosta* occur, suggesting rapid burial without major transportation (Steurbaut and Nolf, 1989).

Similar to the venericards, the turritelline shells are largely intact and mostly devoid of epibionts, arguing against endured transport or exposure at the surface. They are nonetheless slightly worn, probably due to low-energy winnowing in the subtidal distribution channels (Jacobs, 2015). Supply of specimens to the winnowing site by storm events cannot be ruled out, since the coquinas were deposited above the storm wave base. As some modern turritellines are known to also live on tidal flats and low tide beaches (Waite and Allmon, 2013), it is possible that the *H. solanderi* shells were transported to the subtidal channels from local nearby mudflats. This could have resulted in their oxygen isotopic profiles deviating from *V.*
*planicosta*, which is considered to have lived strictly in the subtidal environment represented by the Oedelem Mbr.

The otoliths are the smallest fossils studied (3-5 mm length), and therefore likely most prone to post-mortem sedimentary transport. It is possible that the isotopic composition of the studied otoliths could reflect conditions across a range of microhabitats or nearby environments, including those beyond what was occupied by the turritellines and venericards. On top of this, a major principle in otolith taphonomy is the actualistic observation that otoliths are not digested by predators, but
accumulate in their intestines or are excreted (Nolf, 1985, 1995). It is estimated that the majority of otoliths arriving in a thanatocoenosis are in fact excretion products from free-swimming predatory fish such as sharks, which can migrate over considerable distances. This provides a mechanism through which otoliths from different habitats can be delivered to the subtidal environment where turritellines and carditids live (Nolf, 1995; Vanhove et al., 2012). Importation from deeper, cooler environments could well explain the more positive average $\delta^{18}O$ values of the ophidiid otoliths compared to both congrid
otoliths and turritellines and carditids. Notably, the oxygen isotope profiles of the two ophidiid specimens from the Aalter Sand Fm. at the Scherpenberg locality are strikingly different from those of the ophidiid specimens from the Aalter type locality, with a much larger range (3.4 ‰ vs 1.2 ‰) and slightly more negative values (-2.8 ‰ vs -2.5 ‰). This highlights the large facies diversity, both vertically and laterally, characteristic of the Aalter Sand Fm. (Jacobs and Sevens, 1993). Even though the sedimentary environment was similar between the Aalter and Scherpenberg areas, resulting in the deposition of the
Aalter Sand Fm. at both sites, the two sites were possibly influenced by different water masses or environmental conditions. Alternatively, the post-mortem taphonomic transport pathways of ophidiid otoliths could also have been different between Aalter and Scherpenberg, with for example different hunting grounds for the predatory fish visiting the sites.

Taken together, given the processes involved in the deposition of the coquina beds, it appears that the otolith isotope records could reflect environmental conditions over a wider range of habitats and environments (Fig. 10), due to both sediment
transport and postmortem transport by free-swimming predatory fish. The turritellines are expected to have only been transported over smaller distances, reflecting a more local environment, whereas the large venericards experienced only local winnowing and were not transported over considerable distances.



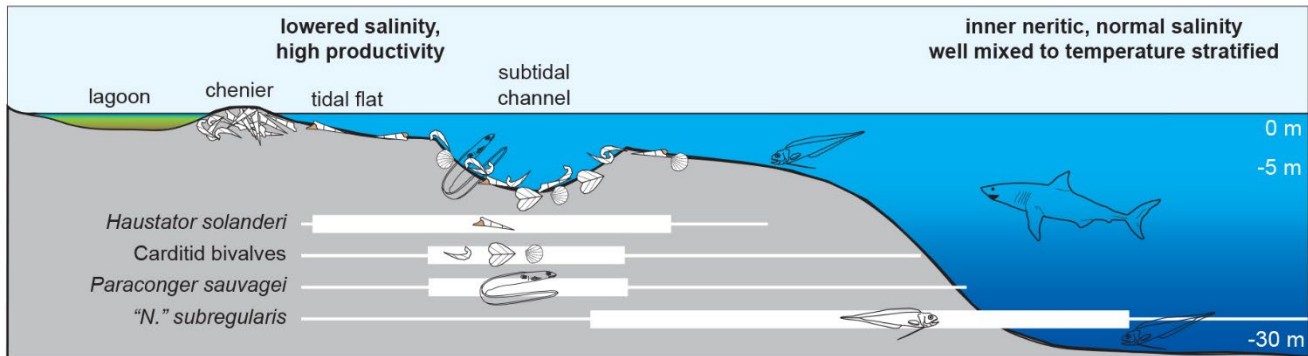


**Figure 10: Simplified paleoenvironmental reconstruction of the coquina layers in the Aalter Sand Fm. and estimated habitat ranges of the organisms used in this study based on species stable isotopes.**

**6.3 Paleobiology and -ecology**

On top of these taphonomic processes, the paleobiology and ecology of the studied species will have played a role in the observed isotopic discrepancies as well, in particular their mobility. Indeed, all studied taxa could move. More mobile taxa may escape to more favorable habitats during some parts of the year, or parts of their life cycle. If they encountered different environmental settings over their life span, such as deeper water or river mouths, this could introduce additional variability in
their isotope profiles.

Among the studied species, carditid bivalves are generally lethargic and move only slowly and infrequently (Yonge, 1969), living just below the surface of the sediment. They are therefore considered most likely to have recorded the in situ intra-annual variation in temperature and/or water composition.

Turritellines live partly buried in sediments with the aperture exposed for filter feeding. Nevertheless, active
movement on the substrate has also been observed and is probably an underestimated characteristic of turritelline behavior (Allmon, 2011). Turritellines are generally considered typical of a subtidal environment (Dominici and Kowalke, 2014). Nevertheless, based on $\delta^{13}$C values and size distribution of *Turritella leucostoma* on the San Felipe tidal flat, where only adults were observed, it was suggested that planktonic larvae of this species settle in deeper settings, while adults move towards more proximal tidal channels and low tide beaches (Allmon, 1988, 2011; Waite and Allmon, 2013), migrating over several hundreds
of meters through their lives. Abundant live adult *T. terebra* individuals have even been collected in estuarine tidal flats (Wu and Richards, 1981). Hence, the potential for turritelline shell growth to record conditions over a wide habitat range is real. Potential mobility of turritellines could explain the variability in their stable oxygen isotope profiles, showing less consistent annual cyclicity than the more sedentary carditid bivalves.

Among the fishes, *Paraconger* is known to live predominantly buried in the substrate, with only the head exposed,
without migrating large distances (Smith, 1989; FishBase, 2023). The otoliths of this species therefore likely primarily record





in situ conditions (e.g. Ivany et al., 2000). Indeed, the taxon mean of *P. sauvagei* from Aalter is statistically indistinguishable from those of the two species of carditid bivalves (*V. planicosta lerichei* and *C. (Arcturellina) sulcata aizyensis*; Welsh test: p<0.01), the least mobile of the studied taxa. The benthopelagic ophidiid '*N.' subregularis*, in contrast, was likely more vagile (FishBase, 2023). As ophidiids today are found over a great range of depths (Vanhove et al., 2012; FishBase, 2023), otoliths

from this taxon could have recorded conditions from a wider spectrum of habitats before coming to rest in the Aalter Sand Fm. The differences in mobility between species can partially account for the differences in taxon means. '*N.' subregularis*, the species with the greatest mobility, exhibits the highest mean $\delta^{18}O$ values, suggesting that it may be less representative of the specific conditions in the immediate Aalter depositional environment, compared to other species. Ophidiid $\delta^{18}O$ values likely reflect more open marine temperatures and $\delta^{18}O_w$. Therefore, even though a taphonomic mechanism through which otoliths

from different settings end up in one death assemblage is likely, short-distance migration to the sedimentation site could have provided additional deviations in $\delta^{18}O$ values of ophidiid otoliths.

   When comparing the isotope profiles of the carditid bivalves, the turritelline gastropods and the congrid fishes, i.e. the taxa that are interpreted to reflect the more local environment of the Aalter Sand Fm., it is notable that, while the average of the 5 percent minimum values differ by less than 0.8 ‰, the average of the 5 percent maximum values differ up to 1.4 ‰

among these taxa (Fig. 8). This difference is most pronounced between the turritelline gastropods and the other taxa. It is possible that the turritellines lived in more proximal settings, somewhat more depleted in winter than the site where the carditids lived and were transported syn/post-mortem to accumulate in tidal channels. Alternatively, in summer, more mobile taxa like the turritelline gastropods could migrate to deeper waters to escape the thermal stress characteristic of intertidal environments (Dong et al., 2022), further reducing their seasonal isotopic range. Nonetheless, it is questionable whether the

very shallow depth gradient in the Belgian part of the sNSB (Jacobs et al., 1991; Jacobs and Sevens, 1993; Vandenberghe et al., 2004) could account for sufficient local depth differences to result in lateral temperature or $\delta^{18}O_{sw}$ variability large enough to result in isotopic differences of up to 1.4 ‰, without invoking significant freshwater input or evaporative conditions. Both evaporative conditions and the significant input of isotopically light freshwater are deemed unlikely, considering the normal salinity conditions indicated by the fauna of the Aalter Sand Fm.

The observed patterns could also be caused by variation in growth rates through the year, with the winter period not fully represented by shell in some of the taxa. Variation in growth rate and shutdown of growth under certain ambient conditions, or in function of their life span, have been documented many times in different types of modern and fossil mollusks (Ullmann et al., 2010; e.g. Versteegh et al., 2010; Schwartzmann et al., 2011; Strauss et al., 2014; De Winter et al., 2018). It is less commonly documented in otoliths, perhaps because it is less common (Wurster and Patterson, 2001; Pilling et al., 2007).

If growth rate reduced significantly during winter, this season will be less represented in the secreted carbonate, resulting in biased specimen and taxon averages and ranges. Potentially, the isotope profile of *V. planicosta*, which encompasses the largest seasonal isotope range, represents the full seasonal cycle, while the isotope profiles of the turritelline gastropods and congrid otoliths only represent part of the seasonal cycle in temperature and sea water isotopic composition.



Unfortunately, rate of growth and growth cessation are difficult to assess in fossil specimens. The shape of the isotope
profiles can provide some clues  (e.g. Goodwin et al., 2003; Ivany, 2012; Judd et al., 2018; De Winter, 2022). Macroscopic
growth increments in our two *V. planicosta* shells co-vary with isotope data, suggesting that the growth banding in venericards
is annual. Even though sampling resolution should be high enough to obtain relatively (quasi)sinusoidal cycles (>10 samples
per annual cycle), most peaks are saw-toothed. Based on cuspate $\delta^{18}$O profiles in fossil venericards from the Eocene, slower
growth in winter months has been inferred in several previous studies (Andreasson and Schmitz, 1996; Purton and Brasier,
1999; Kobashi et al., 2001). Therefore, although the *V. planicosta* shells might represent the most complete seasonal cycle
among the studied taxa, actual $\delta^{18}$O variation could have been even larger. Consequently, the intra-annual ranges recorded in
our specimens should be seen as minimum estimates.

Intra-annual $\delta^{18}$O variations of the turritelline specimens are less straightforward to interpret. Cycles vary in shape
and amplitude, potentially resulting from the more mobile behavior of the turritellines. The results from the ShellChron model
reveal considerable variations in growth rate throughout the lives of the studied specimens, ranging from >1000 µm/day during
growth peaks, to as little as <10 µm/day during winter growth breaks. In several of the studied specimens (e.g. in B11C, B11H,
B11G) some of the winter peaks are sharp and appear to be truncated, which likely shows that time is missing, suggesting that
sampling resolution was not high enough across the increments secreted in winters to capture the full seasonal range. Moreover,
previous studies (e.g. Anderson and Allmon, 2020) have suggested that growth is particularly fast in the first year, followed
by a decline in growth rate in subsequent years. While not immediately evident from the results of this study, a decrease in
growth rate with age would mean that subsequent years require increasingly high sampling resolution to capture the full
seasonal cycle.

The oxygen isotopic profiles of the otoliths of *P. sauvagei* also show a distinct cuspate profile, indicating slower
growth, or even growth cessation, in winter months. The profiles of '*N.*' *subregularis* show less consistent patterns between
the specimens. Given that the otolith records from this species generally represent less than one full year, it is difficult to assess
patterns of growth in these otoliths. Indeed, in organisms that have very short lifespans, the isotopic means and ranges could
be biased, because of incomplete annual cycles.

These observations highlight that life history can introduce additional complexity in interpreting sclerochronological
isotope profiles. Intra-annual records of short-lived species are potentially incomplete and should be interpreted with extra
care. A very short life span might also explain the relatively large range in average $\delta^{18}$O values between the different
*Cyclocardia* shells. To our knowledge, no information is available about the biology, life span or variability in growth rate in
this genus. Hence, these small mollusks may be short-lived, with different individuals having had different spawning seasons
and thus their shell values represent seasonally biased temperatures instead of true mean annual conditions. Further study
would be needed to confirm this.

Of the fish otoliths, the *Paraconger* genus is known to grow older than '*Neobythites*' (Ivany et al., 2003; Vanhove et
al., 2011), which is confirmed here based on the shape of the $\delta^{18}$O profiles, with an estimated lifespan for *P. sauvagei* of up to





2 years, compared to up to one year for '*N.*' *subregularis*. This suggest that intra-annual stable isotope records of *P. sauvagei* are better suited to reconstruct seasonality.

Lifespans of turritellines derived from δ¹⁸O profiles of various modern and fossil species rarely exceed 5 years, with most growth taking place during the first year, followed by a drastic decline in growth rate during the subsequent years (Allmon et al., 1992; Jones and Allmon, 1995; Andreasson and Schmitz, 1996; Jones, 1998; Teusch et al., 2002; Latal et al., 2006; Allmon, 2011; Anderson and Allmon, 2020; Ivany et al., 2018). Our results are in line with a relatively short lifespan for turritellines. The δ¹⁸O profiles of the *H. solanderi* specimens yield estimate ages ranging between 1 and 3.5 years, suggesting that this species lived no more than about 4 years.

Even though we only sampled three years of our *Venericor* specimens for stable isotope analyses, macroscopic growth increments indicate lifespans of 9-13 years for the studied specimens. This is in line with previous studies, which demonstrate lifespans of various *Venericardia* species ranging from 10-30 years (Ivany et al., 2004, 2018). Hence, *V. planicosta* has the longest life span of the studied taxa from the Aalter Sand Fm. In combination with their lethargic behavior and consistent seasonal isotopic signal, *Venericor* likely represents the most reliable of the different types of archives studied here, in terms
of recording environmental conditions at the Aalter paleoenvironment.

### 6.4 Implications for the seasonality in the sNSB

The isotopic profile of *V. planicosta* shells from Aalter show a seasonal range in δ¹⁸O$_{carbonate}$ between -1.5 and -5.3 ‰, reflecting a considerable seasonality in temperature and/or sea water composition. When one would assume a constant sea water oxygen isotopic composition (δ¹⁸O$_w$) throughout the year, a seasonal temperature range could be calculated. For paleotemperature
calculations based on stable oxygen isotopes of aragonitic mollusks, we apply the equation of Grossman and Ku (1986) using a modified version in which δ¹⁸O$_w$ is cast in VSMOW instead of PDB (Goodwin et al., 2001). Given that δ¹⁸O$_w$ is not constrained for the Aalter environment, determining absolute summer and winter temperatures is difficult. While the global average δ¹⁸O$_w$ of the ice-free early Eocene world will have been around ~ -1 ‰ (Zachos et al., 1994; Roberts et al., 2009), water isotope-enabled general circulation models suggest that the partly enclosed early Eocene sNSB region might have been
characterized by slightly more depleted waters, of up to -2 ‰ (Tindall et al., 2010; Zhu et al., 2020). With this estimated δ¹⁸O$_w$, calculated sea water temperatures would correspond to ~18-34 °C, consistent with previously reconstructed mean annual bottom water temperatures of 27-31 °C for the mid to late Ypresian sNSB, using Mg/Ca and clumped isotopes on large benthic foraminifera (Evans et al., 2018; Martens et al., 2022).

If the sea water oxygen isotopic composition was constant throughout the year, the intra-annual range of 3.8 ‰ in *V.*
*planicosta* would imply a range in seasonal water temperatures of ~16 °C. This is considerably more than the modern global average seasonal sea surface temperature range of 8-9 °C at 41°N (Ivany, 2012). It is also more than present-day ranges in the sNSB of about 10-12 °C in a well-mixed water column (Andreasson and Schmitz, 2000; Austin et al., 2006). We therefore consider it possible that the Aalter region was characterized by a seasonal variation in seawater oxygen isotopic composition. Relatively more depleted waters in summer and/or more enriched waters in winter could account for the relatively large



seasonal range recorded in $\delta^{18}O_{carbonate}$. Future study using clumped isotope thermometry would allow us to separate the effects of temperature and seawater isotopic composition on the oxygen isotope composition of carbonates (Eiler, 2007; De Winter et al., 2022; Meckler et al., 2022; Keating-Bitonti et al., 2011), enabling us to constrain the absolute seasonal variability temperature and/or sea water composition during the deposition of the Aalter Sand Formation.

## 6.5 Causes for variability in $\delta^{13}C$

In our dataset, the taxon means of $\delta^{13}C$ values range between -3.9 ‰ (congrid otoliths) and +2.8 ‰ (turritellines), with little overlap between the ranges of the taxa (Fig. 9). The relative sequence of the offsets – turritellines highest values, followed by venericards, ophidiids, and finally congrids – is consistent with most of the limited data available for comparison in literature (e.g. Andreasson and Schmitz, 1996; Ivany et al., 2003; Kobashi et al., 2004; Vanhove et al., 2012), possibly reflecting the respective trophic levels of the studied taxa.

Nevertheless, unraveling the causes of $\delta^{13}C$ variations in biogenic carbonates is known to be considerably more complex than $\delta^{18}O$ (Wefer and Berger, 1991; McConnaughey and Gillikin, 2008). Variations are controlled by the relative contributions of respired (R) and environmental DIC (1-R) and their $\delta^{13}C$ values. The fraction of respired DIC (R) incorporated into biogenic carbonate varies between species and over lifespans as a function of metabolic growth rate (Schwarcz et al., 1998; McConnaughey and Gillikin, 2008). The fraction of respired DIC in shell material is also dependent upon the locus of 640 precipitation within the shell, with the inner shell layer tending to be more depleted in $^{13}C$ than the outer (Ivany et al. 2008). In general, reported R values for modern mollusks are 10 % or less, while those for fish are higher, up to 40 % (Gillikin et al., 2006; Solomon et al., 2006; McConnaughey and Gillikin, 2008; Tohse and Mugiya, 2008). As respired DIC is considerably more depleted than that of ambient seawater (McConnaughey and Gillikin, 2008), a difference in R is a likely explanation for why values of turritellines and venericards are enriched compared to congrids and ophidiids. The marked increase in $\delta^{13}C$ over 645 the life span of the turritellines (Fig 5) could be compatible with an ontogenetic decrease in respired DIC contribution.

Concerning the otoliths, congrids and ophidiids feed at approximately the same trophic level, on plankton and small benthos, including annelid worms, crustaceans and small fish (Takai et al., 2002; FishBase, 2023), which all have tissue $\delta^{13}C$ compositions of around -15 ‰ (Takai et al., 2002). Nevertheless congrid otolith $\delta^{13}C$ is considerably more depleted than ophidiid otolith $\delta^{13}C$. If these fish taxa indeed occupied different habitats, as inferred based on their $\delta^{18}O$ data, the $\delta^{13}C$ offset 650 would support of a more distal habitat for ophidiids, since their $\delta^{13}C$ are closer to $\delta^{13}C$ of open marine DIC.

## 6.6 Comparison with the Egem Sand fauna

While only few sclerochronological studies have been performed on the late Ypresian sNSB, the records from the nearby upper middle Ypresian Egem Mb. (Vanhove et al., 2012) provide an interesting comparison. There, similar taxa were analyzed from the Egem Sand Mb., including otoliths of the congrid *Paraconger papointi* and the ophidiid '*N.*' *subregularis*, and the bivalve 655 *Cyclocardia (Arcturellina) sulcata* (Vanhove et al., 2012). The Egem Mb. was sampled in the Ampe Clay Pit, located 17 km



SW of Aalter. It comprises nearshore to shore-face sand deposits, deposited within the EECO interval, about 2 myr before the deposition of the Aalter Sand Fm (Vandenberghe et al., 2004; Steurbaut, 2015).

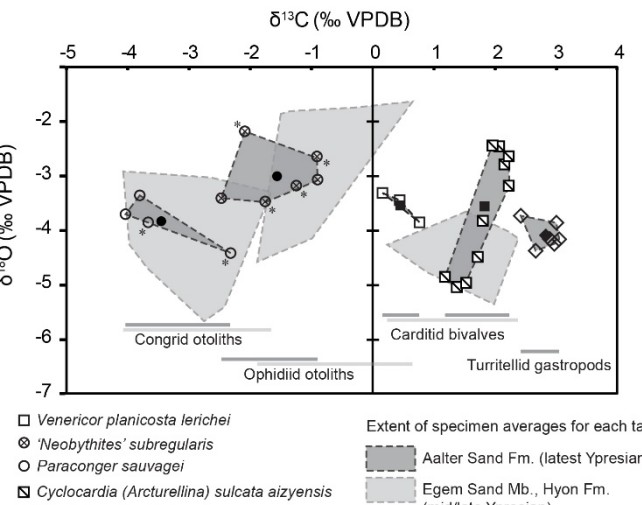

**Figure 11: Cross-plot of specimen average δ18Oc and δ13C values of congrid and ophidiid otoliths, carditid bivalves and turritelline gastropods, based on bulk sampling (bulk line samples or ground whole specimens) or averages of serially sampled data. Specimens marked by an asterisk are from Vanhove et al. (2011). Light grey fields are taxon extents from the inner neritic Egem locality of mid to late Ypresian age (~51 Ma), located 18 km SW of Aalter (Vanhove et al., 2012). The cross-plot shows overall resemblance between both shallow marine sites in the position of otoliths and bivalves on the cross-plot. Taxa can be distinguished from each other on the cross-plot due to the taxon-specific differences in δ13C.**

A cross-plot of specimen averages of δ13C and δ18O data (Fig. 11) reveals consistency between the upper Ypresian Aalter Sand Fm. (this study) and the upper middle Ypresian Egem Mb. (Vanhove et al., 2012). Results from similar taxa show

at least partial overlap in both isotopes. In both the Aalter Sand Fm. and Egem Mb., taxa plot as distinct groups, primarily due to the distinct taxon-specific δ13C signatures of each taxon. In both units, the mean of the specimen averages in δ18O of ophidiids is the most positive of all taxa, corroborating that ophidiids indeed seem to record a different ambient condition compared to the other taxa. For both ophidiids and congrids, the Egem Mb. specimens show a wider spread in δ18O values, possibly related to the more proximal setting of the Egem Sands (Steurbaut, 2006; Martens et al., 2022). A rather significant

difference between the lithostratigraphic units is that the range of the averages of *C. (A.) sulcata aizyensis* from the Aalter Sand Fm. is about double that of *C. (A.) sulcata* from the Egem Sand Mb. While the reason remains unclear, it should be reiterated that their size is very small and little is known about their life span.

The cross-plot demonstrates that by measuring multiple taxa and multiple specimens in a stratigraphic unit, the observed isotopic variation becomes more representative of the 'true' complexity and range of isotopic variation of its

macrofauna.



**Conclusions & recommendations**

Our work has important implications for studies reconstructing intra-annual trends in paleotemperature and seasonality based on skeletal archives from shallow marine settings. Based on the results of this study, several recommendations can be made
for future sclerochronological studies focusing on multitaxon assemblages:

(1) In general, it is crucial to assess the sedimentological and taphonomic processes involved in the deposition of the sedimentary record and fossils within. Many macrofossil-rich deposits are the result of winnowing or transport, causing time averaging within the death assemblage and possibly introducing elements from a variety of different local microhabitats into the assemblage. The least transported fossils, such as bivalves preserved in live position, will yield the most consistent and
representative data. Moreover, the largest fossils are least likely to be transported. Smaller fossils, such as fish otoliths, are more likely to have undergone some transport, both sedimentary as well as intestinal by predatory fish. Otolith isotope records could therefore reflect environmental conditions over a wider range of habitats and environments in comparison to mollusks.

(2) When selecting incrementally mineralized fossils as short-term archives of past climate, the least mobile or most lethargic animals are to be preferred. More motile animals, such as gastropods and fish, can introduce additional variability in
their isotope profiles, as they could reflect a wider range of habitats and environments. Among non-migratory groundfish, congrids likely reflect a more local signal than ophidiids, because they are known to live predominantly buried in the substrate, with only the head exposed, while ophidiids are more vagile benthopelagic fish.

(3) Overall, species with long lifespans provide more robust records of seasonality, compared to species with very short lifespans, as long-lived species are more likely to capture the full seasonal cycle, and provide a more averaged, multi-
year record of seasonality. In this respect, large bivalves can be more long-lived, often >10 years, while gastropods generally life up to 1-5 years. Also congrids generally capture more than one full seasonal cycle. The Paleogene ophidiids from the sNSB, on the other hand, appear to be generally short-lived, hampering accurate assessment of seasonality. At the same time, short-lived species often show faster growth rates, potentially providing higher resolution records, indicating a trade-of between longevity and resolution.

(4) Variation in growth rate and shutdown of growth under certain ambient conditions, during particular seasons, or over ontogeny, has been documented both in fossil mollusks and otoliths, leading to an incomplete representation of the seasonal cycle in their growth increments. In our Paleogene records, congrids, ophidiids and turritellines each show considerable slowdown in growth in parts or their record, possibly not fully capturing the seasonal cycle. Of the studied fossil groups, carditid bivalves appear to record the most complete representation of seasonal cycles. However, the saw-toothed
pattern of their isotope profiles suggests that also the intra-annual ranges recorded in carditid bivalves should be seen as minimum estimates.

(5) In a dynamic coastal environment, accumulation of organisms with different habitat preferences and niches into one thanatocoenosis implies that environmental parameters derived from one taxonomic group, or only one specimen for each taxon, cannot be representative for the isotopic variability in this death assemblage and environment. Examination of multiple



individual specimens spanning several different taxonomic groups considerably increases the likelihood of capturing the full

range of isotopic variation of the environment.

(6) After selecting the most suitable incrementally mineralized fossils, seasonally resolved clumped isotopes analyses

to can separate the effects of temperature and sea water isotopic composition on the oxygen isotope composition of carbonates

and allow reconstruction of the absolute seasonal variability in temperature and sea water composition.




**Appendices**

**Appendix A.** Description of the turritelline and otolith sampling in Aalter, Belgium: V324 ("Aalter Hagepreekstraat")


Date: 01 November 2010

Coordinates (degrees): 51°05'8.51" N 003°27'5.45" E

Address: Hagepreekstraat, Aalter, Belgium

Construction pit for apartment building, samples taken from excavated sands

Constructor: Mevaco, Durmelaan 6, Aalter

2.5-3 meter thick profile visible:

- Sand stone layer of 25 cm thick about 0.8 m below surface, disappears towards the southwest

- Below stone layer: 2m50 brown to green medium sand with shell coquinas, decalcified towards the southwest


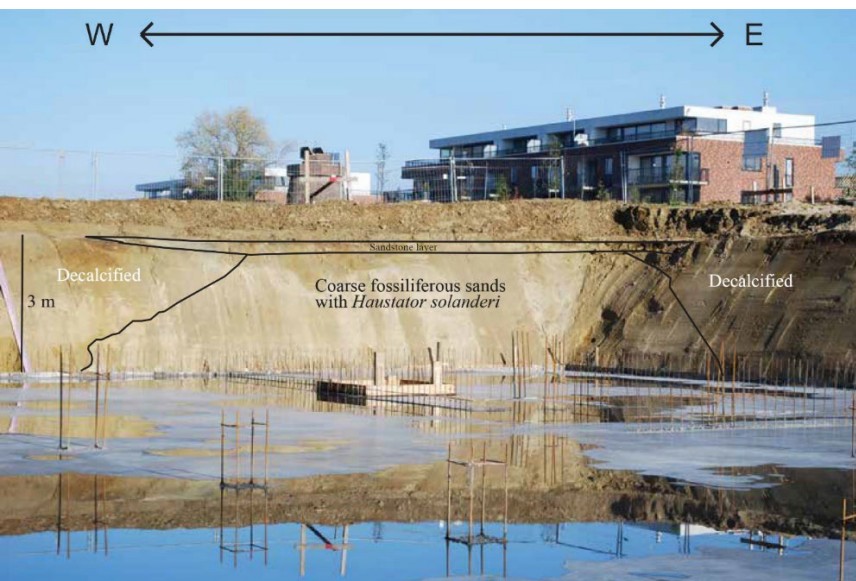

Apppendix A1: Photograph of sampling locality V324 ("Aalter Hagepreekstraat"), taken 01 November 2010. Samples were taken from the 'fossiliferous sands with *Haustator solanderi*', a non-decalcified part of the Aalter Fm. protected from infiltration of meteoric water by a 0.25 m thick sandstone layer.




**Appendix B.** Preservation assessment

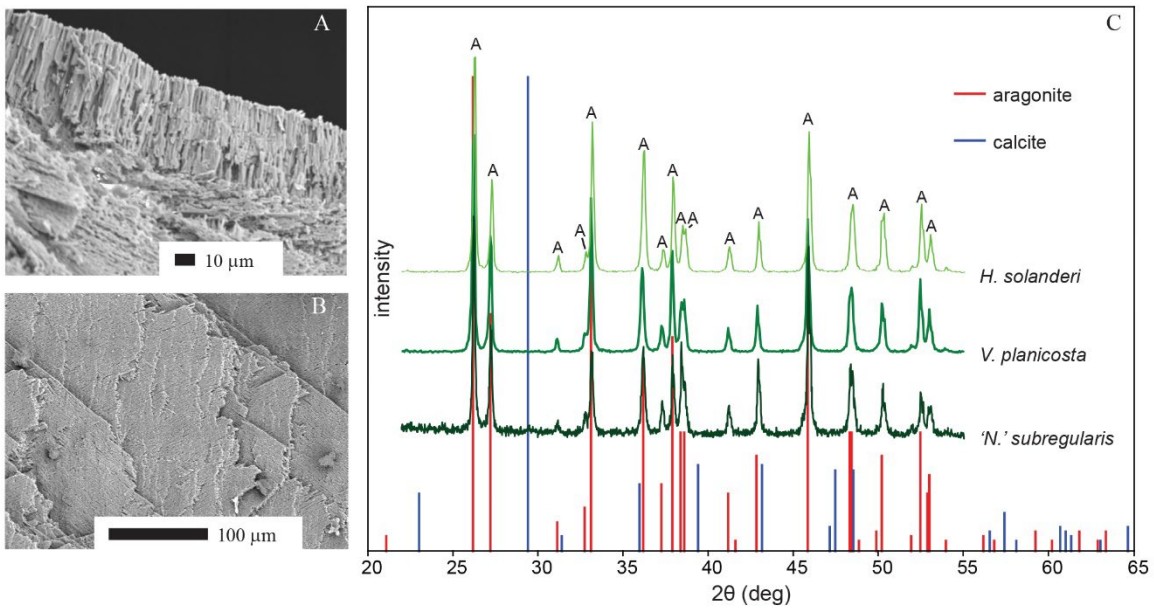

Appendix B1: SEM images of A) *H. solanderi* and B) *V. planicosta lerichei*, both from the Aalter Sand Fm. at Aalter, Belgium, and C) X-ray diffractograms of *H. solanderi*, *V. planicosta lerichei*, and *'N.' subregularis* from Aalter, showing the shells are composed of aragonite. Image A) shows elongated crystals perpendicular to the shell surface. Image B) shows a plated ultrastructure. Structures are similar to those in modern mollusks and indicate original aragonite ultrastructure has been preserved.


**Data availability**

All stable isotope data generated are provided in Supplement S1.


**Author contributions**

JV, DV, ES and RPS conceived the research. DV performed stable isotope analyses. IJ performed ShellChron modelling. PC and LCI provided guidance in stable isotope laboratories. JV and DV wrote the manuscript text; NJDW revised the manuscript text critically.


**Competing interests**

The authors declare that they have no conflict of interest.



**Acknowledgements**

Annelise Folie is thanked for providing access to the INS collections and permission for destructive analyses on loaned specimens of *V. planicosta lerichei*. We are grateful to Peter Stassen for his help with SEM imaging, and Rieko Adriaens for his help with XRD analyses. We thank Herman Nijs, David K. Moss, Lora Wingate and Michael Korntheuer who assisted with sample preparation or stable isotope analysis. This research was funded by the Belgian Federal Science Policy (BELSPO) through FED-tWIN project Prf-2020-038 (MicroPAST), through KU Leuven STG grant 3E211203 to J.V., through a grant by the Agency for Innovation through Science and Technology to D.V. (IWT SB093015) and grants by the Research Foundation Flanders (FWO) to R.P.S, P.C. and E.S. (G.0422.10N) and N.J.W. (12ZB220N). LCI was supported in part by EAR-0719645 from the US National Science Foundation.

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
