# Peer review of "Making sense of variation in sclerochronological stable isotope profiles of mollusks and fish otoliths from the early Eocene southern North Sea Basin"

_EGUsphere, 2024_

## Author Comment (AC1)

Dear editor,

We thank both reviewers for their valuable comments and suggestions on our manuscript. The remarks by Reviewer 1 (Dr. Vickers) were overall positive, and can easily be incorporated in a revised version on the manuscript. Reviewer 2 provided stronger criticism on the manuscript, through arguing that our work is lacking a clear scientific objective and does not provide any new results.

Based on the comments by reviewer 2, we realize that the overall objectives of our research were not sufficiently clear from the manuscript text. While the overarching rationale for studying isotope sclerochemistry of fossil mollusks and otoliths is indeed to reconstruct past climatic conditions, this was not the goal of our research. Instead, given the common use of fossils to reconstruct paleoseasonality, we wanted to investigate isotope variability within one fossil assemblage, i.e. to test whether different fossils from the same assemblage record the same signals or not, and if not, deduce which groups provide the most robust seasonality signal in the Eocene North Sea Basin. Hence, as was correctly assessed by reviewer 2, our study was indeed not designed as a study to reconstruct paleoclimatic conditions during the Eocene.

We now realize that the introduction of our manuscript creates the false impression that our study presents an attempt to understand the environmental, biological, and taphonomical causes for differences in amplitude and mean of intra-annual isotope variability among specimens and species, rather than to assess what these differences *are*. We agree with the reviewer, that to understand some of the causes of this variability, a modern death assemblage could be used. Yet, at the same time, it must be stressed that such an assemblage likely would not incorporate long term taphonomic processes, such as reworking and transport, that affected fossil assemblages. In a revised version of the manuscript, we will stress that the aims of our study are to test whether or not the different fossils from the same assemblage from the Eocene North Sea Basin record the same signals.

Specific replies to detail comments by Reviewer 2:

Reviewer comment: *If the objective is to reconstruct past climate (first sentence of the introduction), the authors should focus on Venericor planicosta, a massive, long-lived, subtidal, non mobile species that provided excellent isotopic records.*

- We are thankful that the reviewer agrees with us that, based on the data presented in our manuscript, *Venericor planicosta* is indeed one of the best candidates for the reconstruction of paleoseasonality in the Eocene North Sea Basin. We will mention this more explicitly in a revised version of our manuscript.

Reviewer comment: *In the paleoenvironment and paleoceanography sections, no open question is presented that could be interesting to tackle with these archives.*

- In contrast to the assumption by the reviewer, the paleoenvironment and paleoceanography sections are designed to provide the reader an understanding of the conditions in the Eocene North Sea Basin. They are not intended to present any research questions. We therefore do not think the lack of questions poses an issue.

Reviewer comment: *L260 : how did you estimate the age (13+) of the specimen?*

- This was done by counting the growth increments. We will indicate this in the revised manuscript.

**Reviewer comment:** *L261: why were years 6, 7, 8 selected? Why not earlier years that would be less affected by growth cessation?*

- The years were selected randomly. At present, we have no independent indication that earlier years in *V. planicosta* are less affected by growth sessation.

**Reviewer comment:** *L390: if htere are less than 40 datapoints, the 5% highest or lowest value are just 1 datapoint.*

- This is correct. For several of the studied specimen, the 5% highest and lowest values were represented by 1 measurement. We will mention this explicitly in a revised version of the manuscript to clarify this to the reader.

**Reviewer comment:** *L630 - 650. continental freshwater input is not mentioned as a possible cause for d13C variations. It is actually one of the main causes in coastal environments.*

- While continental freshwater input is one of the main causes of carbon isotopic variability in biological records from coastal environments, we have several lines of evidence to suggest that freshwater input was limited in the Aalter depositional environment (e.g. the fossil assemblage and sedimentology of the site). These are explicitly addressed in the manuscript text. Moreover, as all but the ophidiids are considered to be derived from the direct surroundings of the depositional site, it seems unlikely that freshwater input, if present at all, would differentially affect bivalves vs gastropods vs congrids. Therefore, the potential freshwater input does not affect our discussion of the differences between the taxa. We already stress this in the manuscript (see section 6.3). In our revised manuscript, we will stress while freshwater input can influence the carbon isotopes profiles in biological records, we consider it unlikely that carbon isotope differences *between* taxa are caused by freshwater input.

**Reviewer comment:** *given the small amount of respired carbon in mollusk shells, the influence of trophic level is not significant. In general most of the text is focused on the respired carbon, and overlooks the cause of variations of DIC d13C which accounts for 90% of shell d13C variability.*

- Both turritellines and venericards are considered to have lived in the direct surroundings of the depositional site (see section 6.2 Taphonomy). It is therefore difficult to envision that the isotopic difference between these taxa is caused by a very local difference in DIC. Please note that most of the text (10 out of 22 lines of the paragraph) is focused on the isotopic difference between the mollusks and the otoliths. Given that fish have a much higher contribution of respired DIC than mollusks, we do consider it likely that this difference in contribution of R can explain the difference between turritellines & venericards versus otoliths.

**Reviewer comment:** *L688-690: what good is measuring the 'true' complexity of isotopic variations if it cannot be deciphered?*

- The point we are trying to make is that only by measuring multiple taxa, one can obtain an understanding of to what extent the different taxa capture the full seasonality. One cannot measure one taxon only and automatically assume that the isotopic variability of this taxon reflects the total seasonality. We will rephrase this sentence to more clearly express this point.

**Reviewer comment:** *L709: "long-lived species are more likely to capture the full seasonal cycle".*
*This is incorrect. The full seasonal cycle is captured if the species lives at least 1 full year without*
*growth cessation. It has nothing to do with the record's length. Robustness may be obtained from*
*multiple specimens.*

-   The full seasonal cycle is captured if the species lives at least 1 full year without growth
    cessation, *and without any random, local variability.* In actuality, sclerochronological records
    often show a large variability, caused by all sorts of local processes, ranging from local
    weather phenomena up to predation attempts. If one has more years within one specimen,
    it becomes easier to identify this random, local variability, and identify the seasonal trends.
    Using more individuals of a species that only lives 1 year introduces additional variability,
    caused by slightly different ages, live positions or growth environment between the
    individuals. At the same time, we realize that truly long-lived species, i.e. those living for
    decades, are likely to incorporate bigger and bigger cessations with age, as fast growth is not
    a major objective. The ideal therefore seems to be with species that live for several years
    only.  In our revised manuscript, we will rephrase to better demonstrate/explain the
    advantages of species that live more than 1 year, while also stressing the limitations of long-
    living taxa.

With best regards,

Johan Vellekoop

Signed on behalf of all the authors